# Improved Cell Selectivity of Pseudin-2 via Substitution in the Leucine-Zipper Motif: In Vitro and In Vivo Antifungal Activity

**DOI:** 10.3390/antibiotics9120921

**Published:** 2020-12-18

**Authors:** Seong-Cheol Park, Heabin Kim, Jin-Young Kim, Hyeonseok Kim, Gang-Won Cheong, Jung Ro Lee, Mi-Kyeong Jang

**Affiliations:** 1Department of Polymer Science and Engineering, Sunchon National University, Suncheon 57922, Korea; schpark9@gnu.ac.kr (S.-C.P.); kimheabin94@naver.com (H.K.); jykim19@scnu.ac.kr (J.-Y.K.); hht95@naver.com (H.K.); 2Division of Applied Life Sciences and Research Institute of Natural Science, Gyeongsang National University, Jinju 52828, Korea; gwcheong@gnu.ac.kr; 3National Institute of Ecology, 1210 Geumgang-ro, Maseo-myeon, Seocheon-gun 33657, Korea

**Keywords:** antimicrobial peptide, leucine-zipper motif, cell selectivity, antifungal action

## Abstract

Several antimicrobial peptides (AMPs) have been discovered, developed, and purified from natural sources and peptide engineering; however, the clinical applications of these AMPs are limited owing to their lack of abundance and side effects related to cytotoxicity, immunogenicity, and hemolytic activity. Accordingly, to improve cell selectivity for pseudin-2, an AMP from *Pseudis paradoxa* skin, in mammalian cells and pathogenic fungi, the sequence of pseudin-2 was modified by alanine or lysine at each position of two amino acids within the leucine-zipper motif. Alanine-substituted variants were highly selective toward fungi over HaCaT and erythrocytes and maintained their antifungal activities and mode of action (membranolysis). However, the antifungal activities of lysine-substituted peptides were reduced, and the compound could penetrate into fungal cells, followed by induction of mitochondrial reactive oxygen species and cell death. In vivo antifungal assays of analogous peptide showed excellent antifungal efficiency in a *Candida tropicalis* skin infection mouse model. Our results demonstrated the usefulness of selective amino acid substitution in the repeated sequence of the leucine-zipper motif for the design of AMPs with potent antimicrobial activities and low toxicity.

## 1. Introduction

The increasing emergence of pathogens showing resistance to conventional drugs and the increasing frequency of microbial infections in immunosuppressed hosts have necessitated the identification of new types of antibiotics showing different mechanisms for the prevention of infection [1,2]. One promising therapeutic strategy for the treatment of multidrug-resistant strains is antimicrobial peptides (AMPs) [3,4,5,6,7]. AMPs are involved in the innate host defense system as a primary barrier against infection in most organisms. Most AMPs share several common characteristics, including membranolytic or intracellular-damaging action, amphipathicity, 10–50 amino acids, and net positive charge; however, some anionic AMPs have been reported [4,5]. Studies of AMPs have investigated their abundance in nature, their mechanisms, and their roles in innate or adaptive immune systems, as well as the effects of amino acid modification and their functions in drug delivery systems [8,9,10,11,12,13,14,15].

Pseudin-2 is extracted from the paradox frog *Pseudis paradoxa* [16]. A previous report demonstrated that it was aggregated in aqueous solution and dissociated into monomers upon binding to the bacterial cell wall component lipopolysaccharide [17]. Pseudin-2 forms an α-helical structure on bacterial membranes, resulting in the formation of pores in both bacterial and fungal membranes. Moreover, pseudin-2 can enter into the cytoplasm and bind to RNA [17]. Its potent antimicrobial activity is related to pore formation, which can reduce membrane potential and release intracellular materials [17]. It can inhibit protein synthesis through electrostatic binding to RNA [17]. Jang et al. designed the Ps-K18 peptide with a Lys substitution for Leu^18^ in Pseudin-2 and showed in vivo antibacterial and anti-inflammatory effects in the septic shock models [18]. Another group reported that a pseudin-2 analog with deletion and substitutions of amino acids exhibited antibacterial activity in a mouse wound model [19].

Although AMPs may have applications as novel antimicrobial reagents, their clinical applications are limited owing to their potential cytotoxicity in mammalian cells. In several studies, leucine-zipper (LZ) sequences in many naturally occurring AMPs have been shown to be responsible for self-aggregated structures and aggregated peptides in an aqueous environment, resulting in cytotoxic effects through permeabilization of the mammalian cell membrane [20]. The LZ motif contains an α-helical conformation with a heptad repeat sequence composed of seven amino acids and facilitates peptide dimerization. When hydrophobic amino acids, such as leucine, isoleucine, and phenylalanine residues, are located in the “a” and “d” positions as shown in Figure 1A, they are lined up on the same side of the helix [21,22]. Asthana et al. demonstrated that the LZ motif in melittin plays a crucial role in its toxic effect rather than antimicrobial activity [21]. As it is known to play important roles in the assembly of DNA-binding proteins [22] or membrane-associated viral fusion proteins [23], they studied the structural integrity and functional activity of melittin by some modifications of an LZ motif. Recently, single or double substitutions in proline residues at the “a” and/or “d” positions of the rationally designed peptide FR-15 have been utilized to reduce the cytotoxicity of the peptide in mammalian cells while maintaining antimicrobial, anti-endotoxin, and anticancer activities [24].

In this study, we aimed to develop AMP analogous to pseudin-2 to improve cell selectivity, increase antifungal activity, and reduce cytotoxicity. To this end, two residues containing leucine, isoleucine, and/or phenylalanine at the “a” and “d” positions in the LZ sequence of pseudin-2 were replaced with alanine or lysine (P2-LZ1–LZ5). The lytic activities of the peptides were investigated using pathogenic fungi, erythrocytes, and mammalian cells. Their secondary structures and membrane-permeable effects in fungal and mammalian membrane-mimicking liposomes were then examined to investigate the relationships between structure and lytic activity, and one of the variants was applied for the analysis of in vivo antifungal activity.

## 2. Results and Discussion

### 2.1. Designation of Pseudin-2 Derivatives Based on the LZ Motif and Structural Orientation

Pseudin-2 self-assembles in aqueous solution [25]. We hypothesized that the high cytotoxicity of this peptide may be related to this aggregate state. As shown in Figure 1A,B, pseudin-2 possessed a heptad repeat sequence, known as the LZ motif. The LZ is an important structural feature mediating the interactions between DNA and eukaryotic transcription regulatory proteins. Amino acids forming a scissors shape are connected at seven sites, and the inner parts are interlocked at sites of leucine residues. In order to improve the cell selectivity of pseudin-2 for fungal and mammalian cells, analogs were designed on a heptad repeat with leucine, isoleucine, and phenylalanine residues at the ‘‘a” and ‘‘d” positions in P1 and P2 lines (Figure 1A,B). The three-dimensional (3D) model structure adopted a typical α-helical structure, and we predicted this sequence to be oriented such that the dimers were facing each other at the P1 and P2 positions (Figure 2). Four double mutated variants, namely, P2-LZ1, P2-LZ2, P2-LZ3, and P2-LZ4 were designed by replacing leucine, isoleucine, and/or phenylalanine residues with an alanine residue at positions 5 and 12, 2 and 9, 12 and 19, and 9 and 16 of pseudin-2, respectively. Alanine was chosen owing to its weakly hydrophobic characteristics. Another analog, P2-LZ5, was designed by substituting phenylalanine and isoleucine residues with lysine residues at positions 9 and 12. Lysine was used here to prevent the formation of a self-assemble structure by allowing electrical repulsion between the opposing residues of the peptides and to alter the secondary structure of the peptide.

Table 1 shows the physicochemical characteristics of pseudin-2 and its derivatives, including the measured molecular mass, hydrophobicity, hydrophobic moment, and net charge. The hydrophobicity values of the analogs were markedly lower than those of pseudin-2. Peptides with high hydrophobicity are likely to be located in fatty acids of lipids inserted in the cell membrane or to be self-assembled, such as adjacent chains of a globular protein. The hydrophobic moment, another important factor mediating the roles of AMPs, is related to the orientation at the surface between polar and nonpolar phases [28,29]. The high hydrophobic moment of pseudin-2 indicates that it is oriented in an amphipathic perpendicular position relative to its axis, e.g., the LZ orientation. We hypothesized that the reduced hydrophobicity and hydrophobic moment of the analogs could be related to the decreased cytotoxicity against mammalian cells.

### 2.2. Cell-Selective Antifungal Activity of the Designed Peptides

#### 2.2.1. In Vitro Antifungal Activity

To elucidate the relationships between the amino acid substitutions and antifungal activity of the derivatives compared with those of pseudin-2, we evaluated the minimum inhibitory concentrations (MICs) in nine fungal strains, including molds and yeasts. As shown in Table 2, pseudin-2 and its derivatives exhibited more potent inhibition of fungal growth in yeast than in mold fungi, except for that in *Fusarium oxysporum*, a phytopathogenic mold. Furthermore, the derivatives had lower MICs at an acidic pH than at a neutral pH. The MICs of P2-LZ1 and P2-LZ3 with substituted amino acids at the P1 position were slightly increased in most fungal cells, whereas those of P2-LZ2 and P2-LZ4, which had substitutions at the P2 position, showed similar or lower MICs compared with pseudin-2. Notably, P2-LZ4 showed potent antifungal activity against pathogenic yeast cells, including *Candida albicans*, *C. krusei*, *C. tropicalis*, and *Trichosporon beigelii*, with MICs ranging from 1.5 to 6 µM at neutral pH; this activity was not observed against *C. parapsilosis,* a yeast fungus that causes serious sepsis or wound infections in immunocompromised patients. We suggest that the difference of MIC values against each fungus is determined by the compositions of the cell wall of each fungus and P2-LZ4 has a high affinity with them. The MICs of P2-LZ5, for which the net charge was increased by lysine substitution, were very different depending on the fungal species.

#### 2.2.2. In Vitro Cytotoxic Effects

2,3-Bis-(2-Methoxy-4-Nitro-5-Sulfophenyl)-2*H*-Tetrazolium-5-Carboxanilide (XTT) assays were performed to evaluate the viability of human HaCaT cells after treatment with the compounds. Although the cytotoxicity of P2-LZ1 was lower than that of pseudin-2, both compounds exhibited significant, concentration-dependent cytotoxic effects against HaCaT cells. Other peptides were not toxic at 256 µM (Figure 3A).

Figure 3B shows the hemolytic effects of peptides at a concentration of 64 µM in erythrocytes. Hemolysis is an important response to intravenous application of peptides. Triton X-100, a detergent, and melittin, a cytotoxic peptide, cause severe hemolysis by inducing hemoglobin release from erythrocytes. However, erythrocytes treated with pseudin-2 or its analogs did not release hemoglobin based on visualization (Figure 3B). The different cytotoxic patterns observed in HaCaT cells and mouse red blood cells (mRBCs) were related to differences in the lipid composition and cell membrane content of the two cell types.

### 2.3. Mode of Action of the Designed Peptides

#### 2.3.1. Secondary and Predicted 3D Structures of the Designed Peptides

To elucidate the structure–activity relationships of peptides, the secondary structures of peptides in fungus-mimic liposomes (phosphatidylcholine (PC)/phosphatidylethanolamine (PE)/phosphatidylinositol (PI)/ergosterol, 5:4:1:2, *w*/*w*/*w*/*w*) were determined by analysis of circular dichroism (CD) spectra (Figure 4). Pseudin-2 and its derivatives adopted an α-helical structure in the presence of vesicles, although their helicities differed. P2-LZ peptides, except for P2-LZ5, induced more helical structures than pseudin-2, whereas P2-LZ5 formed a loose helical structure. Although the CD data of P2-LZ peptides showed a random coil in an aqueous solution (Appendix A), their structures measured in the presence of fungal cell membrane were of helical structure.

#### 2.3.2. Mode of the Antifungal Actions of the Designed Peptides

Although *C. albicans* is the major fungus causing candidiasis, we used *C. tropicalis* as a model fungus in the mechanism study because pseudin-2 and analogs showed the best activity in *C. tropicalis* among the candida species (Table 2).

To ascertain the cellular distributions of the designed peptides, the localization of rhodamine-labeled peptides in *C. tropicalis* cells was observed by confocal laser-scanning microscopy (CLSM). Melittin, which is derived from honeybee venom and forms pores in the plasma membrane of eukaryotic cells [30], and pseudin-2 [17] are membrane-active peptides. Both rhodamine-labeled peptides were more accumulated on the surface than in the cytosol of fungal cells (Figure 5). As shown in our report [17], pseudin-2 enters into the cytoplasm through a hole made by itself and is binds to nucleic acids. We suggested that P2-LZ1, -LZ2, -LZ3, and -LZ4 would have a similar mechanism of action to pseudin-2. On the other hand, P2-LZ5 was detected inside the cells, but not in the nucleus. This result suggests that the antifungal mechanisms of P2-LZ5 may be different from other peptides.

In general, studies of the mode of action of AMPs and peptides begin by measuring the membrane permeability of microbes in the presence of the peptides. Therefore, we assessed changes in the fluorescence of SYTOX Green, an impermeable nuclear and chromosome counterstain in live cells, over time (Figure 6). Melittin and pseudin-2 induced gradual, concentration-dependent increases in fluorescence intensity, although the degree of fluorescence intensity differed. Changes in the fluorescence of LZ-P peptides were similar to those of pseudin-2, except for P2-LZ5. Furthermore, although the time of maximum fluorescence emission differed depending on the treatment concentration, the fluorescence intensity was highest at 9–12 min in the presence of 64 µM peptides, indicating that the antifungal mechanism involved membranolysis. The fluorescence intensity in the presence of P2-LZ5 was much lower than that in the presence of the other peptides. Typically, cell-permeable AMPs show no significant increases in fluorescence intensity when the peptide concentration is increased [31,32], and the membrane permeability of the dye is low when the peptide is applied at concentrations above the MIC value. Therefore, we believe that these peptides could penetrate into the cytosol of fungal cells.

Analysis of the degree of dye leakage, indicative of the ability of a peptide to disrupt the membrane, showed that melittin and pseudin-2 caused a significant release of calcein from PC/PE/PI/ergosterol vesicles as a fungus-mimic membrane (Figure 7A). Melittin, pseudin-2, P2-LZ1, P2-LZ2, and P2-LZ4 peptides induced 84.1%, 80.1%, 81.8%, 77.0%, and 80.5% leakage of calcein, respectively at a molar ratio of 0.05, whereas the leakage percentages of P2-LZ3 and P2-LZ5 were 48.3% and 33.8%, respectively, at the same molar ratio (Figure 7A). In addition, melittin and pseudin-2 induced significant dye leakage from a mammalian-mimic membrane composed of PC/CH/sphingomyelin (SM) vesicles, whereas P2-LZ peptides caused less than 20% calcein efflux at a peptide/lipid molar ratio of 0.1, except for P2-LZ1 (Figure 7B). These results were consistent with the observed toxic effects of the peptides in HaCaT cells. The antimicrobial and cytotoxic effects of pseudin-2 and its analogs may be related to the cell membrane composition, and the cell selectivity of these peptides may be associated with the orientation of the peptides in aqueous solution owing to changes in hydrophobicity and the hydrophobic moment.

To investigate the modes of action of P2-LZ peptides in fungal cells, we assessed the morphology of *C. tropicalis* cells in the presence of peptides by scanning electron microscopy (SEM). As shown in Figure 8, control cells exhibited smooth surfaces without damage, whereas melittin formed large holes in the surface of fungal cells. In the presence of pseudin-2, the fungal cell surface showed many holes and torn parts, and the cell morphology was wrinkled; similar results were observed for P2-LZ1 and P2-LZ3. In contrast, P2-LZ2 and P2-LZ4 induced cell swelling as well as holes in the cell surface. These results suggested that the antifungal mechanisms of P2-LZ1 and P2-LZ3 were different from those of P2-LZ2 and P2-LZ4. In contrast, *C. tropicalis* cells were relatively large, wrinkled, and swollen following the addition of P2-LZ5. In particular, the surface pores in these cells appeared to be created by pushing into the cell from the outside. This shape was not observed for the other peptide treatments.

### 2.4. Mitochondrial Reactive Oxygen Species (ROS) Generation in Response to P2-LZ5

To investigate the antifungal mechanism of P2-LZ5, the production of mitochondrial superoxide (SOX) in fungal cells was measured in the presence of peptide. The mitochondrial membrane of eukaryotic cells is anionic, similar to the bacterial cell membrane. When cationic peptides enter the cells, they can target mitochondria and cause disruption of membrane potential, release of cytochrome *c*, or generation of ROS, leading to cell apoptosis [33]. As shown in Figure 9, cytometry peaks indicated that MitoSOX Red fluorescence was significantly increased in the presence of P2-LZ5, as demonstrated by the right shift in the fluorescence emission. While pseudin-2 showed a little right shift and others did not. As shown in the results of other mechanisms, they did not induce intracellular ROS because they can act rapidly on fungal cell membranes and kill fungal cells. Figure 10 indicates that P2-LZ5 had different antifungal mechanisms compared with others. We propose that the P2-LZ5 peptides entered into the fungal cells can easily bind to the mitochondrial membrane with a negative charge via electrostatic interaction, resulting in the induction of destabilizing membrane potential and mitochondrial ROS.

### 2.5. In Vivo Antifungal Effects

*C. tropicalis* induces various infectious diseases, including oropharyngeal candidiasis, oral thrush, vulvovaginal candidiasis, angular cheilitis, pulmonary candidiasis, and gastrointestinal candidiasis, depending on the specific tissue or organ that it colonizes. Among the analog peptides, P2-LZ4 has the best antifungal activity and has no cytotoxicity, therefore it was selected in the animal experiment. P2-LZ5, showing a different mechanism, increased net charge due to the increase of lysine residues, resulted in a significant decrease of its antifungal activity under PBS condition (high salt). We evaluated the in vivo fungicidal activities of pseudin-2 and P2-LZ4 peptides in a mouse model of *C. tropicalis* skin infection. At 24 h after subcutaneous fungal infection, mice were treated with phosphate-buffered saline (PBS), pseudin-2, or P2-LZ4. As shown in Figure 10A, control mice treated with PBS exhibited obvious swelling and redness in the dorsal skin, whereas psuedin-2 treated mice showed reduced swelling and redness. Surprisingly, no obvious skin lesions appeared in the presence of P2-LZ4. Next, histological evaluation of skin tissues was performed using Hematoxylin and Eosin (H&E) staining (Figure 10B). Notably, *C. tropicalis*-infected mice showed markedly increased inflammatory cell numbers, inflammatory cell infiltration, and tissue necrosis (Figure 10B(a2,b2)) compared with PBS-treated mice (Figure 10B(a1,b1)). Although pseudin-2 treated mice were able to recover from histological lesions (Figure 10B(a3,b3)), skin tissues from mice treated with P2-LZ4 were nearly identical to those of the PBS-treated mice (Figure 10B(a4,b4)).

## 3. Materials and Methods

### 3.1. Materials

Triton X-100, Calcein, cholesterol (CH), eosin Y, ergosterol, hematoxylin, hexamethyldisilazane (HMDS), and XTT were purchased from Sigma-Aldrich Co. (St. Louis, MO, USA). PE (from the brain), PC (from eggs), sphingomyelin (SM, from egg), PI (from bovine liver), and a mini-extruder kit were from Avanti Polar Lipids (Alabaster, AL, USA). Ethyl cyanohydroxyiminoacetate (oxyma) and 9‑fluorenylmethoxycarbonyl (Fmoc) amino acids were obtained from CEM Co. (Matthews, NC, USA) N,N′-diisopropylcarbodiimide (DIC) was acquired from Tokyo Chemical Industry Co., Ltd. (Tokyo, Japan). SYTOX Green, 5/6-carboxy-tetramethyl-rhodamine succinimidyl ester (NHS-rhodamine), and MitoSOX Red were purchased from Molecular Probes (Eugene, OR, USA). All other reagents were of analytical grade.

### 3.2. Fungal Strains and Growth

*Aspergillus flavus* (KCTC 6905), *A. fumigatus* (KCTC 6145), *Fusarium moniliforme* (KCTC 6149), *F. oxysporum* (KCTC 16909), *C. albicans* (KCTC 7270), *C. krusei* (CCARM 14017), *C. parapsilosis* (CCARM 14016), *C. tropicalis* (KCTC 7221), and *T. beigelii* (KCTC 7707) were obtained from the Korea Collection for Type Cultures (KCTC) and Culture Collection of Antimicrobial Resistant Microbes (CCARM). Mold and yeast fungi were pre-grown on potato dextrose (PD) and yeast extract peptone dextrose (YPD) agars, respectively.

### 3.3. Peptide Synthesis

Microwave-assisted peptide synthesis (Discover Bio, CEM Co., Matthews, NC, USA) was used to synthesize the peptides. The amidated peptides were obtained by Rink amide 4-methylbenzhydrylamine resin. Fmoc amino acids were coupled using microwave heating in the presence of DIC and Oxyma in dimethylformamide (DMF) and Fmoc deprotection was processed using 20% piperidine in DMF. After the final coupling and deprotecting steps, the resin was washed with dichloromethane and air-dried. The peptides were cleaved from the resin, using trifluoroacetic acid (TFA)/triisopropylsilane/DiH_2_O (95:2.5:2.5, *v*/*v*/*v*) for 2 h at room temperature, followed by precipitation and washing with ice-cold diethyl, and dried under a vacuum. The crude peptides were purified using a ZORBAX PrepHT Eclipse C_18_ preparative column (21.2 × 150 mm, 5-μm) on a Shimadzu high-performance liquid chromatography (HPLC) system. The purity of the purified peptides was more than 98% in analytic HPLC. Their molecular masses were measured using a matrix-assisted laser desorption ionization mass spectrometer (MALDI II; Kratos Analytical Ltd., Manchester, UK) [17].

### 3.4. Prediction of the 3D Structure

The 3D structural prediction of each peptide was modeled in PEP-FOLD online server (https://bioserv.rpbs.univ-paris-diderot.fr/services/PEP-FOLD3/) [26,27] and the best-fitted structures were rendered with PyMol (PyMOL Molecular Graphics System, v2.0 Schrödinger, LLC, New York, NY, USA).

### 3.5. Antifungal Assay

Antifungal activity of peptides was performed by microtiter plate assays. Fungal conidia from four-day-old molds were collected using 0.08% Triton X-100 (*v*/*v*), and pre-cultivated yeast cells were adjusted to 2 × 10^4^ spores/mL in 10 mM sodium phosphate buffer (SP, pH 7.2) or 10 mM 2-(N-morpholino)ethanesulfonic acid (MES) buffer (pH 5.5) supplemented with 20% culture medium, followed by addition to peptide solutions (0.25, 0.5, 1, 1.5, 2, 4, 6, 8, 12, 16, 32, 48, and 64 µM) in 96-well microtiter plates. After incubation for 36 h at 28 °C, the hyphal growth of conidia and the proliferation of yeast cells were observed under a light microscope. All assays were performed in triplicate. The MIC values were defined as the lowest concentration of samples that reduced fungal germination (for mold) or cell proliferation (for yeast) by more than 90% [34].

### 3.6. Cytotoxic and Hemolytic Assay

Cell viability was assessed using sodium XTT assays. HaCaT (human immortalized keratinocytes) cells were cultured in Dulbecco’s modified Eagle medium (DMEM; ThermoFisher Scientific, Gibco, Waltman, MA, USA) supplemented with antibiotic-antimycotic (ThermoFisher Scientific, Gibco) and 10% fetal bovine serum (FBS; ThermoFisher Scientific, Gibco) at 37 °C in a humidified chamber containing 5% CO_2_. The cells were seeded at 5 × 10^4^ cells/mL in flat-bottom 96-well microtiter plates in triplicate. Twenty-four hours later, cells were treated with two-fold serial dilutions of peptides (ranging from 256 to 4 µM) in medium, and the cells were further incubated for 24 h. Next, activated-XTT solution was added to each well and the plates were additionally incubated for 4 h. Absorbance in each well was measured at wavelengths of 480 and 650 nm using a microtiter SpectraMax M5 reader (Molecular Devices, Sunnyvale, CA, USA). Triton X-100 (0.1% *v*/*v*) and DMEM were used as controls for survival [31].

Mouse blood was collected into sodium heparin-coated tubes (BD Vacutainer; BD Diagnostics, Oxford, UK). The mRBCs were obtained by centrifugation (800× *g*, 10 min) and washing with PBS. The washed mRBCs were added to a final concentration of 8% (*v*/*v*) in 64 µM peptide solution, followed by incubation for 2 h at 37 °C. After centrifugation, each tube was digitally recorded.

### 3.7. CD Analysis

A Jasco J-810 spectropolarimeter (Jasco, MD, USA) was used to analyze the secondary structure of each peptide in artificial liposomes. A peptide solution mixed with PC/PE/PI/ergosterol (5:4:1:2, *w*/*w*/*w*/*w*) liposomes at a 1:10 molar ratio (30 µM peptide/300 µM lipid) was injected into a 0.1-cm path-length quartz cell. At least five scans were acquired and averaged from 190 to 250 nm. Samples were allowed to equilibrate to 25 °C prior to CD measurement. The experiments were run at 50 nm/min with 1 nm data intervals. The base line was adjusted with only liposome. Mean residue ellipticities ((θ), deg·cm^2^ dmol^−1^) were calculated by following equation [31].
(θ) = θ_obs_ /10·*l*·*c*
where θ_obs_ is the measured signal (ellipticity) in millidegrees, *l* is the optical path-length of the cell in cm, and *c* is the concentration of peptide in mol/L [mean residue molar concentration: *c* = number of residues in the constructed of peptide × the molar concentration of the peptide].

### 3.8. CLSM

To conjugate the fluorescent probe at the N-terminus of the peptide, NHS-rhodamine was incubated with peptide solutions at a 1:1 molar ratio in PBS (pH 7.2) at room temperature for 1 h. Rhodamine-labeled peptides were purified using a C_18_ column on an HPLC system. Peptide solutions were used with the mixtures of rhodamine-labeled peptides and rhodamine-free peptides (1:9, *w*/*w*) to minimize rhodamine effects. Peptides were then added to 200 μL of *C. tropicalis* cell suspensions at the appropriate MIC. After incubation for 1 h, the cells were pelleted by centrifugation at 3000× *g* for 5 min, washed with ice-cold PBS, and fixed with 2% glutaraldehyde (*v*/*v*) in PBS. The cellular distribution of rhodamine-labeled peptides was then examined using an inverted LSM510 laser scanning microscope (Carl Zeiss, Gőttingen, Germany). The 543 nm light from a helium neon laser was directed at a UV/488/543/633 beam splitter. Images were recorded digitally in a 512 × 512 pixel format [17].

### 3.9. SYTOX Green Uptake

*C. tropicalis* cells pre-grown at 28 °C were washed and suspended (2 × 10^6^ cells/mL) in SP buffer (pH 7.2). The cells were pre-incubated with SYTOX Green (final concentration of 0.5 μM) for 15 min in the dark, followed by adding two-fold serial dilutions of peptides at concentrations of 1 to 64 µM. The fluorescence intensity was monitored for 30 min at 485 (Ex) and 520 (Em) nm using a microtiter SpectraMax M5 reader (Molecular Devices, San Jose, CA, USA) [35].

### 3.10. Calcein Leakage from Artificial Vesicles

The permeability of peptides in calcein-entrapped artificial liposomes was assayed by measuring calcein leakage. The desired mixtures of lipids (PC/PE/PI/ergosterol, 5:4:1:2 and PC/CH/SM, 2:1:1, weight ratio) were dissolved in chloroform at 10 mg/mL, dried in a glass tube under nitrogen, and then lyophilized overnight to remove residual solvent. The dried lipid films were resuspended in 1 mL of dye solution (80 mM calcein, 10 mM HEPES buffer, pH 7.2) at 50 °C and vortexed, followed by freeze-thawing for nine cycles and extrusion with polycarbonate filters (0.2-μm pore-size) using an Avanti Mini-Extruder (Avanti Polar Lipids Inc., Alabaster, AL, USA). Free calcein was removed using PD-10 desalting columns (GE Healthcare Life Science, Seoul, Korea). Calcein-entrapped vesicles of 20 μM lipids were incubated with peptides at concentrations of 0.002, 0.005, 0.04, 0.02, 0.05, and 0.1 molar ratio (peptide/lipid) and the fluorescence was assessed at 480 (Ex) and 520 (Em) nm using a microtiter SpectraMax M5 reader (Molecular Devices). Complete (100%) release was achieved by the addition of 0.03% Triton X-100 (*v*/*v*) [17].

### 3.11. SEM

Peptides were incubated with precultivated *C. tropicalis* cells (1 × 10^6^ cells/mL) at the MIC value for 2 h. Cell suspensions were fixed with 2% glutaraldehyde (*v*/*v*) in 0.1 M HEPES buffer (pH 7.4) at room temperature. The PBS-washed cells were subsequently postfixed in 1% osmium tetroxide (*w*/*v*; Electron Microscopy Sciences, Hatfield, PA, USA) for 1 h. The washed cells were dehydrated in OTTIX Shaper (Diapath S.p.A, Bergamo, Italy), followed by chemical-drying using HMDS. Samples were sputter-coated with gold-palladium and observed under SEM (JSM-7100F; JEOL, Ltd., Tokyo, Japan).

### 3.12. Measurement of Mitochondrial SOX

Mitochondrial SOX was measured using MitoSOX Red probe. *C. tropicalis* cells (2 × 10^4^ spores/mL) were treated with samples at the appropriate MIC for 8 h and then stained with MitoSOX Red according to the manufacturer’s instructions. The stained cells were acquired by flow cytometry (Attune NxT Acoustic Focusing Cytometer; ThermoFisher Scientific Co., Eugene, OR, USA) at 561 nm by 10,000 total events.

### 3.13. In Vivo Animal Experiment

All animal experiments and procedures were performed with the approval of the Institutional Animal Care and Use Committee (IACUC) of Sunchon National University, Korea (SCNU IACUC-2019-10). BALB/C female mice (six weeks old) were obtained from Koatech Co. (Pyongtaek, Gyeonggido, Korea) and divided into three groups (*n* = 4): control, pseudin-2, and P2-LZ4 groups. The skin on the back of each mouse was shaved 1 h before infection, and the mice were anesthetized by inhalation of 5% (induction) and 2% (maintenance) isoflurane in pure oxygen. Mice were then subjected to subcutaneous infection with 100 µL *C. tropicalis* (10^7^ cells/mL) by injection into the dorsum of each mouse. At 24 h after infection, 100 µL PBS, pseudin-2 (0.1 mg/mL), or P2-LZ4 (0.1 mg/mL) was injected into the same site. At four days after peptide injection, mice were euthanized by CO_2_ inhalation, and the infected skin tissues were excised, fixed in 4% paraformaldehyde, embedded in paraffin, stained with H&E, and observed by light microscopy.

## 4. Conclusions

In summary, we successfully designed pseudin-2 derivatives to reduce cytotoxic effects based on the LZ hypothesis. The designed P2-LZ peptides formed α-helical structures, similar to pseudin-2. Among the analogs, P2-LZ2 and P2-LZ4 peptides showed excellent broad-spectrum antifungal activities against pathogenic molds and yeasts and were nontoxic against mammalian cells. The peptides P2-LZ1–LZ4, which were mutated by substitution with alanine, exhibited antifungal activity via membranolytic action, whereas the antifungal mechanism of P2-LZ5, which was mutated by substitution with lysine, was related to cell penetration and ROS generation. In addition, P2-LZ4 showed noticeable antifungal effects in vivo in a *C. tropicalis* skin infection mouse model. Our results could provide important insights into the design of cell-selective AMPs via amino acid substitution in the LZ motif.

## Figures and Tables

**Figure 1 antibiotics-09-00921-f001:**
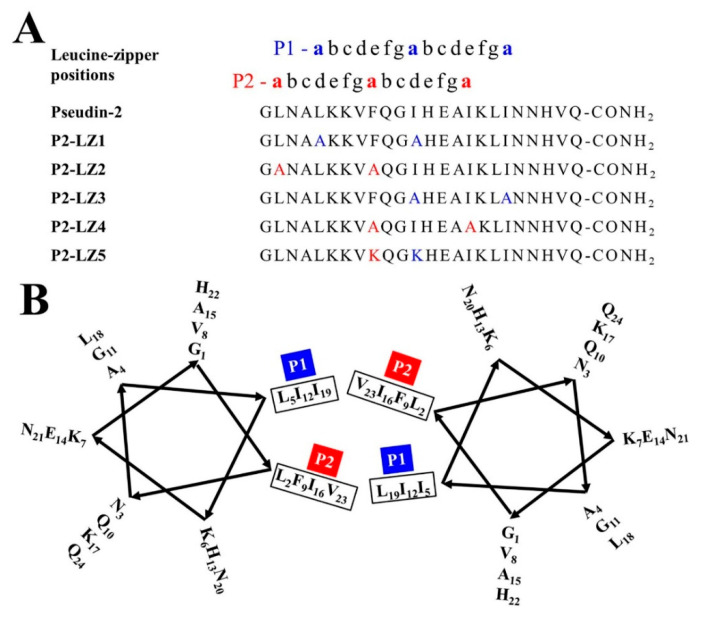
Amino acid substitutions in the leucine-zipper motif of pseudin-2. (**A**) Amino acid sequences of the designated peptides. Amino acids at position “a” are indicated in red and blue colors. (**B**) Helical wheel projection of psuedin-2 in the leucine-zipper orientation.

**Figure 2 antibiotics-09-00921-f002:**
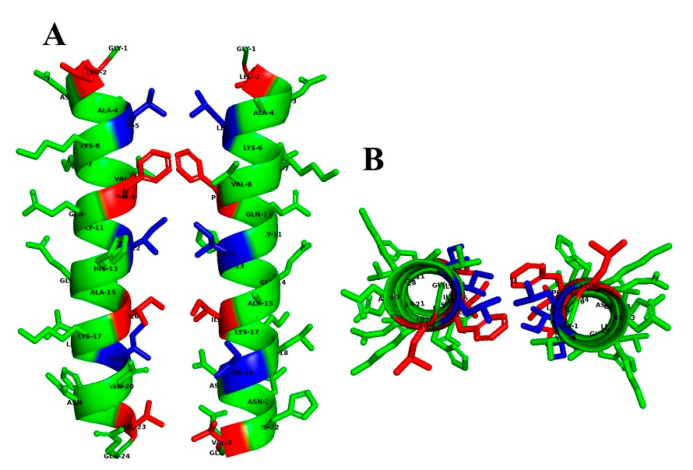
3D model of pseudin-2 in an aqueous solution. Side (**A**) and top (**B**) views of pseudin-2 in the leucine-zipper orientation determined via 3D structure modeling using the PEP-FOLD server (https://bioserv.rpbs.univ-paris-diderot.fr/services/PEP-FOLD3/) [26,27]. Blue and red colors indicate amino acids that could be substituted at P1 and P2 positions, respectively.

**Figure 3 antibiotics-09-00921-f003:**
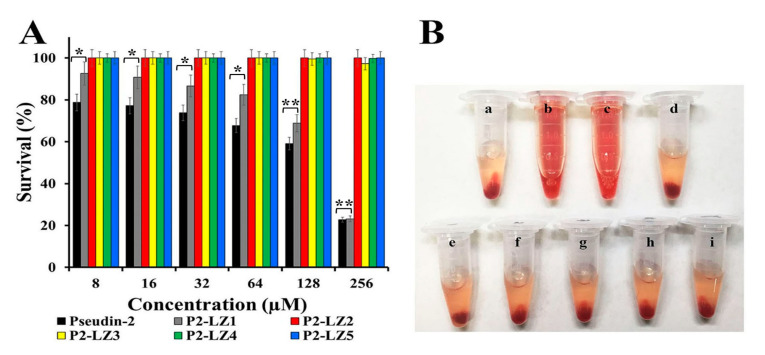
Cytotoxic and hemolytic effects of pseudin-2 and P2-LZ peptides in HaCaT cells (**A**) and mouse erythrocytes (**B**). (**A**) After 24 h incubation of HaCaT cells in the presence of pseudin-2 or P2-LZ peptides, cell proliferation was measured using Alama Blue assays. Data are expressed as the means ± standard deviation from three independent experiments (* *p* < 0.05, ** *p* < 0.01). (**B**) Mouse red blood cells were incubated for 4 h in the absence (**a**) or presence of 0.1% Triton X-100 (**b**), melittin (64 µM) (**c**), pseudin-2 (64 µM) (**d**), P2-LZ1 (64 µM) (**e**), P2-LZ2 (64 µM) (**f**), P2-LZ3 (64 µM) (**g**), P2-LZ4 (64 µM) (**h**), or P2-LZ5 (64 µM) (**i**).

**Figure 4 antibiotics-09-00921-f004:**
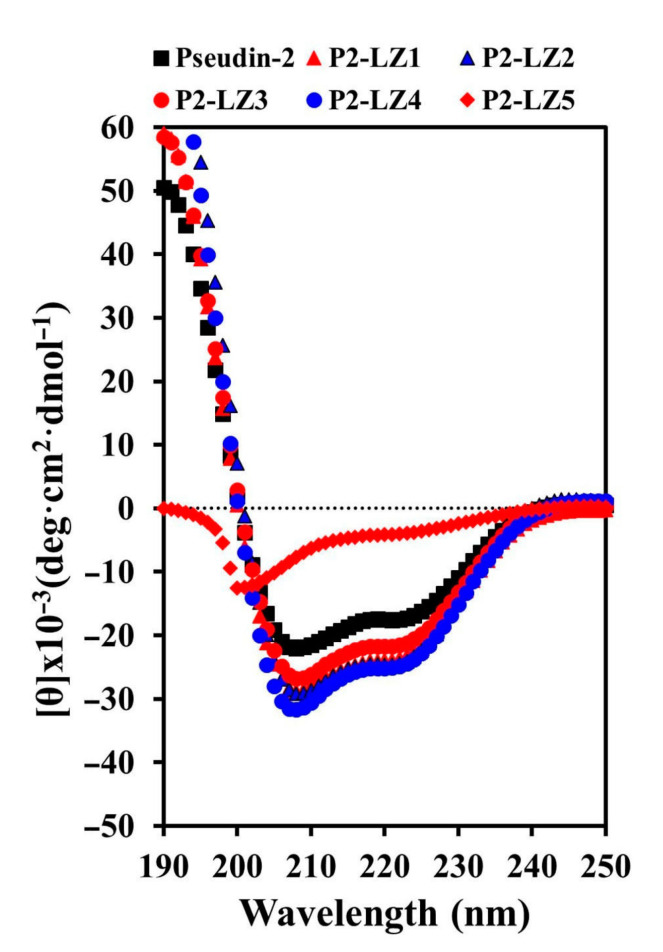
CD spectra of pseudin-2 and P2-LZ peptides. Far-UV CD spectra of peptides (30 µM) were obtained in PC/PE/PI/ergosterol (5:4:1:2, *w*/*w*/*w*/*w*) liposomes.

**Figure 5 antibiotics-09-00921-f005:**
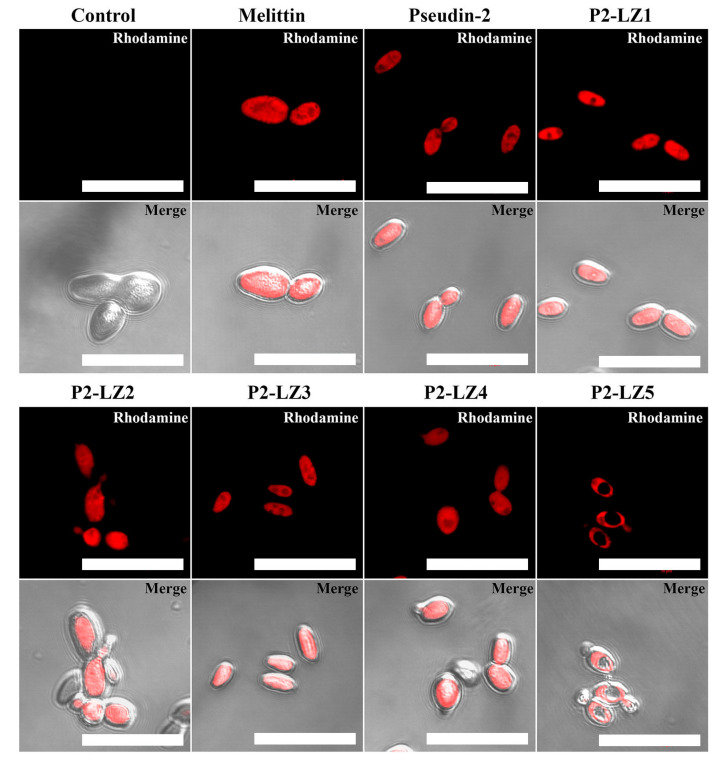
Cellular distributions of rhodamine-labeled pseudin-2 and P2-LZ peptides in *C. tropicalis* cells. Bar is 20 µm.

**Figure 6 antibiotics-09-00921-f006:**
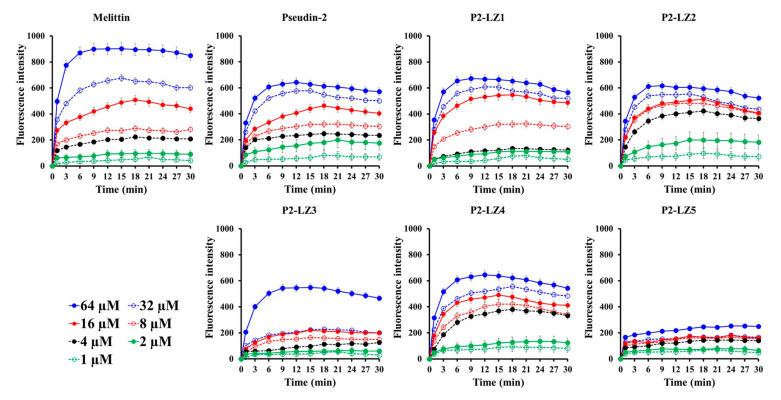
Membrane permeability of pseudin-2 and P2-LZ peptides in *C. tropicalis* cells. Data are expressed as the means ± standard deviation from three independent experiments.

**Figure 7 antibiotics-09-00921-f007:**
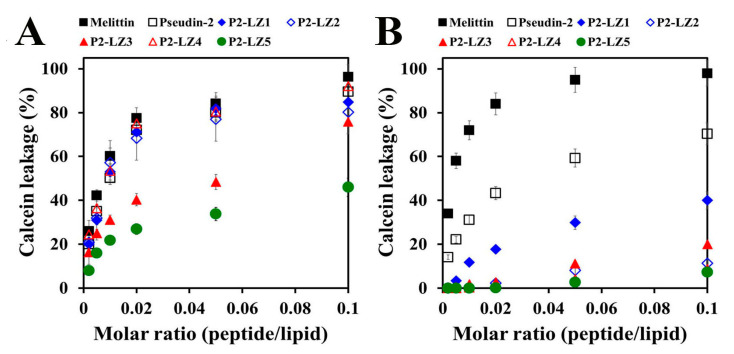
Membranolytic action of pseudin-2 and P2-LZ peptides in artificial liposomes. Calcein leakage from PC/PE/PI/ergosterol (5:4:1:2, *w*/*w*/*w*/*w*) (**A**) and PC/CH/SM (2:1:1, *w*/*w*/*w*) (**B**) liposomes after the addition of the peptides at the indicated molar ratios. Data are expressed as the means ± standard deviation from three independent experiments.

**Figure 8 antibiotics-09-00921-f008:**
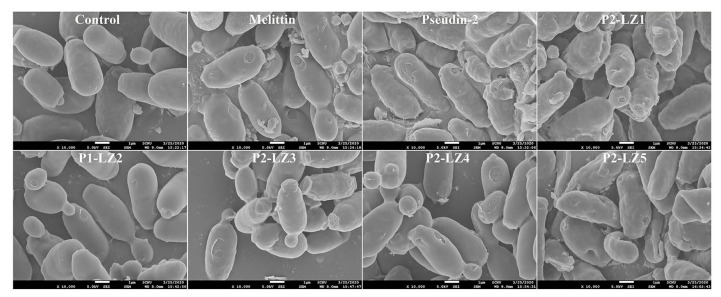
Morphological changes of *C. tropicalis* cells in the presence of pseudin-2 and P2-LZ peptides.

**Figure 9 antibiotics-09-00921-f009:**
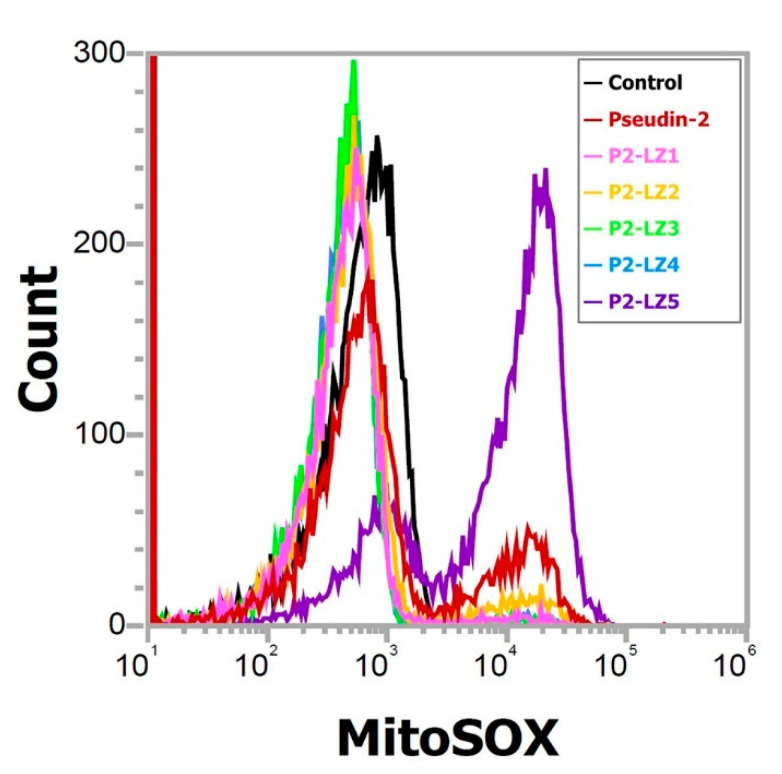
ROS generation induced by peptides in *C. tropicalis* mitochondria.

**Figure 10 antibiotics-09-00921-f010:**
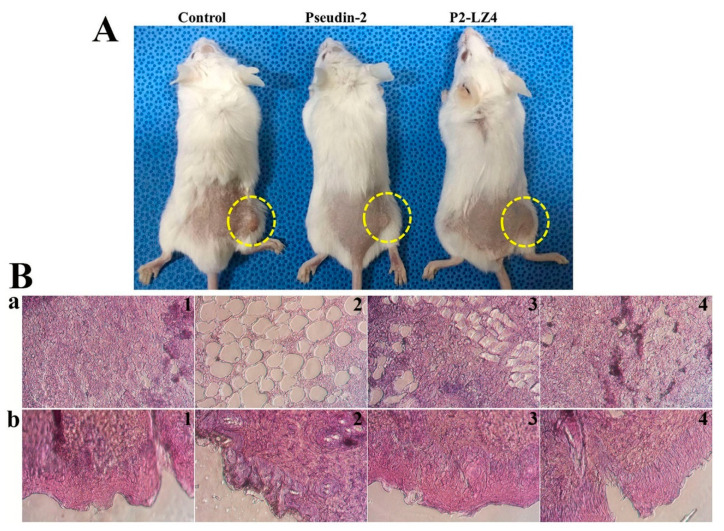
In vivo antifungal activity of pseudin-2 and P2-LZ4 peptides in *C. tropicalis*-infected mice. (**A**) At 24 h after subcutaneous fungal infection, pseudin-2 and P2-LZ4 peptides were subcutaneously injected, and mice were monitored for seven days. (**B**) Paraffin-embedded skin tissues were sectioned and stained with hematoxylin and eosin. Subcutaneous (**a**) and epidermal tissues were observed under a microscope (magnification: 400×). (**1**): PBS, (**2**): *C. tropicalis*, (**3**): *C. tropicalis* with pseudin-2, (**4**): *C. tropicalis* with P2-LZ4.

**Table 1 antibiotics-09-00921-t001:** Physicochemical characteristics of pseudin-2 and its derivatives.

Peptide	Calculated Mass	Observed Mass ^1^	*H* ^2^	*μH* ^3^	RT (min) ^4^	Net Charge
Pseudin-2	2682.5	2685.8	0.407	0.547	45.974	+3
P2-LZ1	2598.4	2601.8	0.287	0.430	43.920	+3
P2-LZ2	2564.5	2567.6	0.287	0.514	38.558	+3
P2-LZ3	2598.4	2601.4	0.283	0.428	43.549	+3
P2-LZ4	2564.5	2567.3	0.283	0.474	44.849	+3
P2-LZ5	2678.6	2681.2	0.175	0.382	33.234	+5

^1^ Mass data shown in Appendix A. ^2^
*H*: Hydrophobicity. ^3^
*μH*: Hydrophobic moment. *H* and *μH* were calculated using the HeliQuest online server. ^4^ RT is retention time on HPLC system (Appendix A).

**Table 2 antibiotics-09-00921-t002:** Antifungal activities of pseudin-2 and its derivatives against mold and yeast cells.

Fungi	MIC (µM)						
	Melittin	Pseudin-2	P2-LZ1	P2-LZ2	P2-LZ3	P2-LZ4	P2-LZ5
Mold							
*A. flavus*	6 (4)	64 (16)	64 (12)	32 (12)	64 (32)	32 (16)	16 (>64)
*A. fumigatus*	6 (4)	64 (24)	64 (16)	24 (16)	64 (16)	32 (16)	16 (>64)
*F. moniliforme*	32 (32)	32 (64)	48 (48)	24 (16)	64 (32)	32 (32)	64 (>64)
*F. oxysporum*	64 (32)	4 (16)	4 (12)	2 (16)	4 (16)	2 (16)	6 (16)
Yeast							
*C. albicans*	32 (8)	12 (32)	8 (32)	4 (32)	8 (16)	4 (8)	32 (64)
*C. krusei*	8 (4)	8 (8)	8 (8)	6 (4)	8 (8)	6 (8)	48 (16)
*C. parapsilosis*	16 (8)	>64 (>64)	> 64 (>64)	48 (64)	>64 (>64)	48 (>64)	>64 (>64)
*C. tropicalis*	16 (8)	4 (2)	4 (2)	2 (1.5)	2 (2)	2 (1)	8 (16)
*T. beigelii*	2 (2)	2 (2)	2 (2)	2 (2)	6 (2)	1.5 (1)	8 (32)

Antifungal assays were performed in 10 mM sodium phosphate buffer (pH 7.2) and 10 mM MES buffer (pH 5.5) supplemented with culture medium. The values in parentheses are MICs under acidic conditions.

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
