# Peer review of "Improved Cell Selectivity of Pseudin-2 via Substitution in the Leucine-Zipper Motif: In Vitro and In Vivo Antifungal Activity"

_antibiotics, 2020, doi:10.3390/antibiotics9120921_

Round 1

Reviewer 1 Report

Result and discussion

How can you explain the differences, that you highlighted in the MIC values, in the various fungal strains? Please add a comment in the text.

In Figure 2 the negative control, cells without any treatment, is missing. Please add the missing data.

Statistical analysis is missing in all the experiments, please perform it and add it in the Figures.

I suggest dividing Figure 6 in 2 different figures in order to enhance the comprehension.

Please add a more detailed explanation of the results obtained with MitoSOX.

Why the in vivo test was performed using P2-LZ4 and not other peptides, for example P2-LZ5 that is the peptide with a different behaviour. Please describe better the choice in the text.

As indicated for Figure 6, I suggest dividing Figure 8 in 2 different figures.

Material and methods

I suggest moving Materials and methods section before the results and discussion one. I think that in this way the comprehension of the paper will be better.

In the Materials and methods section there is not any reference, please add them.

Please put the full name of the technique in the titles of the paragraphs in Material and methods section, for example: 3.7. CD Analysis and 3.8. CLSM.

Author Response

Thank you for your time and efforts in the review of our manuscript. We found that your comments are very constructive and insightful and that the helpful criticism has enabled us to make a better and more informative manuscript. We hope that our revision has improved the paper a level of your satisfaction.

Below we have provided a detailed response to each comment, including:

Our responses to the comment – italicized

Corrections in manuscript – red-colored words

Result and discussion

Q1) How can you explain the differences, that you highlighted in the MIC values, in the various fungal strains? Please add a comment in the text.

A1) We suggest that the difference of MIC values against each fungus is determined by the compositions of the cell wall of each fungus and P2-LZ4 has high affinity with them.

Q2) In Figure 2 the negative control, cells without any treatment, is missing. Please add the missing data.

A2) We calculates the survival percentage by following parameters:

Specific absorbance filter: 475nm, Non-Specific absorbance filter: 660nm

(The XTT-specific absorbance is measured at 450 nm. The 660 nm absorbance reading is used to eliminate the background signal contributed by cell debris or other non-specific absorbance.)

Specific Absorbance = [Abs475nm(Test) – Abs475nm(Blank)] – Abs660nm(Test)

The percentage of the survival cells was calculated using the following formula:

Survival % = [(sample Abs)/ (negative control Abs)] ×100.

Since the above calculation contains negative control Abs, it is not shown in Figure 4A.

In Figure 2B, negative control is a tube of Figure 4B-a.

Q3) Statistical analysis is missing in all the experiments, please perform it and add it in the Figures.

A3) Statistical analysis was performed on the results of Figure 4A, but we believe that the results of other experiments do not require statistical significance.

Q4) I suggest dividing Figure 6 in 2 different figures in order to enhance the comprehension.

A4) We divided to Figure 8 and 9.

Q5) Please add a more detailed explanation of the results obtained with MitoSOX.

A5) We added a more detailed explanation in text.

Q6) Why the in vivo test was performed using P2-LZ4 and not other peptides, for example P2-LZ5 that is the peptide with a different behaviour. Please describe better the choice in the text.

A6) Among the analogue peptides, P2-LZ4 has the best antifungal activity and has no cytotoxicity, therefore it was selected in animal experiment. P2-LZ5, showing a different mechanism, increased net charge due to the increase of lysine residues, resulted in a significant decrease of its antifungal activity under PBS condition (high salt).

Q7) As indicated for Figure 6, I suggest dividing Figure 8 in 2 different figures.

A7) Thank you for your suggestion. But in order to describe Figures 8A and B sequentially, we want to keep the form of figure 8.

Material and methods

Q8) I suggest moving Materials and methods section before the results and discussion one. I think that in this way the comprehension of the paper will be better.

A8) We cannot change because the order is required by this journal.

Q9) in the Materials and methods section there is not any reference, please add them.

Please put the full name of the technique in the titles of the paragraphs in Material and methods section, for example: 3.7. CD Analysis and 3.8. CLSM.

A9) References were added. Full names are already mentioned in section 2, so we expressed to abbreviations.

Reviewer 2 Report

The manuscript describes the antifungal action of pseudin-2 derivatives designed to improve cell selectivity. Overall, the studies are well designed, and the authors performed a well-suited array of methodologies to characterize the peptide action. The results are interesting and detailed. I would suggest some improvements before the paper is ready for publication:

Introduction:

Please ensure that all claims are justified by providing references. For example, sentences ending at page 2 line 45; page 2 line 50 among others.

Please refer to Figure 1 when detailing the structure of pseudin, that would help readers to better understand the terminology. Also, please explain further the terminology “a”-“d”, this is not clear neither from the text nor from the figure.

Please justify why the manuscript is focused in the antifungal activity. Is this peptide active against bacteria? Why is antifungal activity so relevant for these peptides?

Results:

Page 4, lines 117-119: this claim is also true for melittin. Please add more discussion here.

Page 4, line 133: HaCaT cells are not a standard model for cytotoxicity testing. The use of HaCaT cells is not justified in this paper. Results should be compared with other more standard cells lines used for cytotoxicity experiments, such as HEK.

Page 5, line 148: The authors must show the absorbance measurements for hemolysis. As a general guideline, authors should never use “data not show” statements. If data is deemed not relevant enough to be included in the main article, results should be included in Supplementary Information.

Page 6, lines 158-162: Based on CD results, models and experiments do not entirely agree. This raises the following question: Are models accurate enough? The authors should discuss on these differences and detail the limitations of the models used.

Page 6, lines 170-177: Why authors use C. tropicalis instead of C. albicans? Use and selection of model organisms should be discussed in the paper. The authors also state that “… both rhodamine-labelled peptides accumulate on the surface of fungal cells”. This claim is not supported by the data. Based on Figure 4, it is not possible to discard that peptides also accumulate in the cytoplasm of cells. If authors use CLSM, cross-sectional areas could be imaged for better detail. Also, colocalization with specific markers is also advisable.

Page 8, lines 196-198: please provide evidence for that claim.

Page 9, line 210: authors compare data at 0.05 ratio and conclude that P2-LZ3 has a lower percentage of leakage. However, the same peptide has similar leakage at higher ratios. As I understand it, the authors suggest that P2-LZ3 has a different behavior, which is not taking into account all data points.

Page 10, lines 225-226: “… representing ‘barrel-stave’ and ‘toroidal pore2 models, respectively”. This is, in my opinion, an overstatement. Rigorous structural analysis should be performed to conclude mechanisms of action.

Figures:

Figure 1: A-D terminology is not clear. Names in captions B and C are not legible.

Figure 3A: Authors should display the CD spectra of peptides and liposomes alone. Are PepFold models built considering a hydrophobic environment? If not, models built assuming a water solvent cannot be compared to CD spectra in liposomes.

Figure 4: What does control mean? Please increase bar size.

Materials and methods:

  1. Please provide HPLC and MS spectra for all peptides synthesized as Supplementary Information.
  2. Antifungal assay (page 12, line 300): specify peptide concentration range and how serial dilutions were prepared.
  3. Cytotoxic and hemolytic assay (page 12, line 310): specify peptide concentration range and how serial dilutions were prepared.
  4. Cytotoxic and hemolytic assay (page 13, line 324): what does “measured digitally” means? Please specify.
  5. CD analysis (page 13, line 325): Please specify peptide and liposome concentrations. Also provide additional details on how spectra were recorded: scan velocity, etc. How experiments were controlled by the presence of liposomes?
  6. CLSM (page 13, line 331): Authors should provide information on the labelling efficiency. Please specify peptide and rhodamine dye concentrations and the buffer used in the reaction and provide HPLC spectrum of labelled peptides as Supplementary Information. Did the rhodamine labelling affect the biological activity of the peptide? At least MIC or toxicity for labelled peptides should be tested to ensure no labelling effects. More information on confocal conditions are required: excitation and emission wavelengths, …
  7. SYTOX Green Uptake (page 13, line 340): Please provide volumes for cell suspension and sytox at preincubation and how peptide serial dilutions were performed.
  8. Calcein leakage (page 13, line 345): Please provide the weight of dry lipids and volume of dye solution used. Specify peptide concentration range and how serial dilutions were prepared.
  9. SE (page 13, line 355): Please explain why tropicallis cells were incubated with peptides at a 4xMIC concentration.
  10. Mitochondrial SOX (pages 13-14, line 363): Please specify cell density and peptide concentration range and how serial dilutions were prepared. Also describe flow cytometry with more details, including parameter adjustment.
  11. In vivo experiments (page 14, line 368): How many animals were used in the experiment? How were divided in the three different groups? Please specify why a concentration of 0.1mg/mL was used for animal experiments.

Author Response

The manuscript describes the antifungal action of pseudin-2 derivatives designed to improve cell selectivity. Overall, the studies are well designed, and the authors performed a well-suited array of methodologies to characterize the peptide action. The results are interesting and detailed. I would suggest some improvements before the paper is ready for publication:

Thank you for your time and efforts in the review of our manuscript. We found that your comments are very constructive and insightful and that the helpful criticism has enabled us to make a better and more informative manuscript. We hope that our revision has improved the paper a level of your satisfaction.

Below we have provided a detailed response to each comment, including:

Our responses to the comment – italicized

Corrections in manuscript – red-colored words

Introduction:

Q1) Please ensure that all claims are justified by providing references. For example, sentences ending at page 2 line 45; page 2 line 50 among others.

A1) We added references and some texts were corrected.

Q2) Please refer to Figure 1 when detailing the structure of pseudin, that would help readers to better understand the terminology. Also, please explain further the terminology “a”-“d”, this is not clear neither from the text nor from the figure.

A2) “a”-“d” was referred to “as showed in Figure 1A”

Q3) Please justify why the manuscript is focused in the antifungal activity. Is this peptide active against bacteria? Why is antifungal activity so relevant for these peptides?

 A3) Analogues designed in this study had an antibacterial activity against pathogenic bacteria, however, we focused the cell selectivity between fungal and mammalian cells because they are eukaryotic cells and have a neutral charge (zwitterion) in cell membrane surface.

Results:

Q4) Page 4, lines 117-119: this claim is also true for melittin. Please add more discussion here.

A4) The pattern of antifungal activity of melittin on the mold and yeast was different, and it did not have a histidine residue. Also, melittin was used only as a control for antifungal activity. The main purpose of this study is to compare pseudin-2 and analogues.

Q5) Page 4, line 133: HaCaT cells are not a standard model for cytotoxicity testing. The use of HaCaT cells is not justified in this paper. Results should be compared with other more standard cells lines used for cytotoxicity experiments, such as HEK.

  1. A) Many studies have been used HaCaT cell in cytotoxicity assay. As you know, HEK is a human embryonic kidney cell and HaCaT is a human immortalized keratinocytes. Our animal experiments used a skin infection model, so we think that it is correct to perform cytotoxicity assay in HaCaT cells.

Q6) Page 5, line 148: The authors must show the absorbance measurements for hemolysis. As a general guideline, authors should never use “data not show” statements. If data is deemed not relevant enough to be included in the main article, results should be included in Supplementary Information.

  1. A) Sentence of “or absorbance measurement (data not shown)” was deleted. We just want to show visually identifiable data in this manuscript.

Q7) Page 6, lines 158-162: Based on CD results, models and experiments do not entirely agree. This raises the following question: Are models accurate enough? The authors should discuss on these differences and detail the limitations of the models used.

  1. A) Because CD results are the measured structures in the fungal cell membrane and 3D model shows the predicted structures in an aqueous solution, the CD data and the model structures does not exactly match.

Q8) Page 6, lines 170-177: Why authors use C. tropicalis instead of C. albicans? Use and selection of model organisms should be discussed in the paper. The authors also state that “… both rhodamine-labelled peptides accumulate on the surface of fungal cells”. This claim is not supported by the data. Based on Figure 4, it is not possible to discard that peptides also accumulate in the cytoplasm of cells. If authors use CLSM, cross-sectional areas could be imaged for better detail. Also, colocalization with specific markers is also advisable.

A8) Although C. albicans is the major fungus causing candidasis, we used C. tropicalis as a model fungus in the mechanism study because pseudin-2 and analogues showed the best activity in C. tropicalis among the candida species (Table 2).

We agree with the reviewer's opinion, but believe that the peptide is mainly accumulated on the surface of fungal cells. In addition, we think that cellular distribution of peptides is able to accurately distinguish through serial-section examination in CLSM, but it can be clearly seen that it acts on the cell membrane in other mechanism experiments. So we corrected to "more accumulated on the surface than in the cytosol of fungal cells".

Q9) Page 8, lines 196-198: please provide evidence for that claim.

A9) We added reference 29 and 30 as evidences.

Q10) Page 9, line 210: authors compare data at 0.05 ratio and conclude that P2-LZ3 has a lower percentage of leakage. However, the same peptide has similar leakage at higher ratios. As I understand it, the authors suggest that P2-LZ3 has a different behavior, which is not taking into account all data points.

A10) In the calcein leakage results (figure 5), there was a significant difference at the concentrations of 32 µM and 64 µM of P2-LZ3. So, we compared peptides at a concentration of 0.5.

Q11) Page 10, lines 225-226: “… representing ‘barrel-stave’ and ‘toroidal pore2 models, respectively”. This is, in my opinion, an overstatement. Rigorous structural analysis should be performed to conclude mechanisms of action.

 A11) This is just our suggestion, but this sentence has been deleted according to the opinion of the reviewer.

Figures:

Q12) Figure 1: A-D terminology is not clear. Names in captions B and C are not legible.

A12) We divided figure 1 into figures 1 and 2.

Q13) Figure 3A: Authors should display the CD spectra of peptides and liposomes alone. Are PepFold models built considering a hydrophobic environment? If not, models built assuming a water solvent cannot be compared to CD spectra in liposomes.

A13) The 3D model was separated into figure 2.

Q14) Figure 4: What does control mean? Please increase bar size.

A14) Control is non-treated cell and figure was edited.

Materials and methods:

Q15) Please provide HPLC and MS spectra for all peptides synthesized as Supplementary Information.

A15) We understand the reviewer's opinion. But until now, we have never received a review requesting HPLC and MS data while submitting a paper. We confirmed the purity and molecular weight of the isolated peptide and performed an experiment.

Q16) Antifungal assay (page 12, line 300): specify peptide concentration range and how serial dilutions were prepared.

A16) We performed experiments at concentrations of 0.25, 0.5, 1, 1.5, 2, 4, 6, 8, 12, 16, 32, 48, and 64 µM to determine the clear MIC values at low concentrations. So we corrected to “addition to peptide solutions (0.25, 0.5, 1, 1.5, 2, 4, 6, 8, 12, 16, 32, 48, and 64 µM)”

Q17) Cytotoxic and hemolytic assay (page 12, line 310): specify peptide concentration range and how serial dilutions were prepared.

A17) We corrected.

Q18) Cytotoxic and hemolytic assay (page 13, line 324): what does “measured digitally” means? Please specify.

A18) We corrected to “each tube was digitally recorded”.

Q19) CD analysis (page 13, line 325): Please specify peptide and liposome concentrations. Also provide additional details on how spectra were recorded: scan velocity, etc. How experiments were controlled by the presence of liposomes?

A19) We added more information.

Q20) CLSM (page 13, line 331): Authors should provide information on the labelling efficiency. Please specify peptide and rhodamine dye concentrations and the buffer used in the reaction and provide HPLC spectrum of labelled peptides as Supplementary Information. Did the rhodamine labelling affect the biological activity of the peptide? At least MIC or toxicity for labelled peptides should be tested to ensure no labelling effects. More information on confocal conditions are required: excitation and emission wavelengths, …

A20) All comments of reviewers have already been checked prior to performing experiments. We added more information in text.

Q21) SYTOX Green Uptake (page 13, line 340): Please provide volumes for cell suspension and sytox at preincubation and how peptide serial dilutions were performed.

A21) In this experiment, volume is not critical, but it is important to adjust the final concentration of STOX Green to 0.5 µM. We added to “two-fold serial dilutions of peptides”.

Q22) Calcein leakage (page 13, line 345): Please provide the weight of dry lipids and volume of dye solution used. Specify peptide concentration range and how serial dilutions were prepared.

A22) We corrected.

Q23) SE (page 13, line 355): Please explain why tropicallis cells were incubated with peptides at a 4xMIC concentration.

A23) The number of fungal cells used in the antifungal activity assay and in the SEM experiment is different. So, we confirmed 4-times increment of the MIC values of each peptide in 1 × 106 cells/mL and we performed this experiment.

Q24) Mitochondrial SOX (pages 13-14, line 363): Please specify cell density and peptide concentration range and how serial dilutions were prepared. Also describe flow cytometry with more details, including parameter adjustment.

A24) We added cell density in text and the treated concentration of the peptide has already been described as the MIC value. Parameters for flow cytometry were added.

Q25) In vivo experiments (page 14, line 368): How many animals were used in the experiment? How were divided in the three different groups? Please specify why a concentration of 0.1mg/mL was used for animal experiments.

A25) Four mice per group were used. Three groups were divided into control (C. tropicalis+PBS), pseudin-2 (C. tropicalis+pseudin-2), and P2-LZ4 (C. tropicalis+P2-LZ4). Before the animal experiment, we determined the concentration that kills 90% at the infectious concentration of the fungus (Before the animal experiment, we determined the concentration of P2-LZ4 that kills 90% at the infectious density of the fungus (107 cells/mL), and peptides were treated at 0.1 mg/mL.

Round 2

Reviewer 1 Report

The requests made by this Reviewer were answered.

I think that the manuscript could be accepted in this form.

Author Response

English language and style are fine/minor spell check required

The requests made by this Reviewer were answered.

I think that the manuscript could be accepted in this form.

A1) Thanks for your review. Our manuscript was edited by native English speaker.

Reviewer 2 Report

The authors have answered almost all my comments. However, there are several points that were not addressed and I cannot recommend publication until these are addressed in the manuscript, mainly the ones concerning CD spectra and HPLC and MS data.

More specifically:

In response to:

The pattern of antifungal activity of melittin on the mold and yeast was different, and it did not have a histidine residue. Also, melittin was used only as a control for antifungal activity. The main purpose of this study is to compare pseudin-2 and analogues.

This is precisely the point here. The authors attribute the difference between neutral and acidic pH to the presence of His residues. However, melittin shows the same behavior and has no histidines in its sequence. Hence, the presence of absence of His residues is not enough to explain the MIC reduction in acidic conditions. I understand that the main purpose of this study is to compare pseudin-2 and its analogues but this is not incompatible with describing the results with precision and detail.

In response to:

We agree with the reviewer's opinion, but believe that the peptide is mainly accumulated on the surface of fungal cells. In addition, we think that cellular distribution of peptides is able to accurately distinguish through serial-section examination in CLSM, but it can be clearly seen that it acts on the cell membrane in other mechanism experiments. So we corrected to "more accumulated on the surface than in the cytosol of fungal cells".

I understand that the authors may have their theories but these need to be supported by data. Otherwise the authors can only speculate or guess.

In response to:

The 3D model was separated into figure 2.

This does not answer the request: Authors should display the CD spectra of peptides and liposomes alone.

This is important to understand how peptides change its structure upon membrane binding and to assess how reliable the models are.

In response to:

We understand the reviewer's opinion. But until now, we have never received a review requesting HPLC and MS data while submitting a paper. We confirmed the purity and molecular weight of the isolated peptide and performed an experiment.

I also understand the author's opinion but providing HPLC and MS data in manuscripts is a matter of good laboratory and publication practices. I cannot recommend publication unless the authors provide that information.

In response to:

The number of fungal cells used in the antifungal activity assay and in the SEM experiment is different. So, we confirmed 4-times increment of the MIC values of each peptide in 1 × 106 cells/mL and we performed this experiment.

I can understand that, for some reason, authors decided to use a higher cell concentration and use a different MIC, but they need to explain why they did that. Why increase the cell concentration? If the MIC is increased, the authors should explain that they repeated the MIC assay and determined the MIC to be 4x the reported value.

Author Response

Review #2

The authors have answered almost all my comments. However, there are several points that were not addressed and I cannot recommend publication until these are addressed in the manuscript, mainly the ones concerning CD spectra and HPLC and MS data.

More specifically:

Q1) In response to:

The pattern of antifungal activity of melittin on the mold and yeast was different, and it did not have a histidine residue. Also, melittin was used only as a control for antifungal activity. The main purpose of this study is to compare pseudin-2 and analogues.

This is precisely the point here. The authors attribute the difference between neutral and acidic pH to the presence of His residues. However, melittin shows the same behavior and has no histidines in its sequence. Hence, the presence of absence of His residues is not enough to explain the MIC reduction in acidic conditions. I understand that the main purpose of this study is to compare pseudin-2 and its analogues but this is not incompatible with describing the results with precision and detail.

A1) I agree with your opinion, but we have already presented the structural differences between melittin and pseudin-2 in our previous study (reference 17). The quaternary structure of melittin (Proline in the middle of the sequence forms a kink structure in secondary structure, and it is formed to a tetramer by the concentration of peptide and salt in the quaternary structure) is different from that of pseudin-2. There are many factors that can be affected by pH, so we think it is difficult to accurately describe the increased antifungal activity of melittin. Since pseudin-2 increases cations with acidic pH, we have only discussed "suggestion". By your comment, we removed this part from the text because the explanation for histidine is ambiguous.

Q2) In response to:

We agree with the reviewer's opinion, but believe that the peptide is mainly accumulated on the surface of fungal cells. In addition, we think that cellular distribution of peptides is able to accurately distinguish through serial-section examination in CLSM, but it can be clearly seen that it acts on the cell membrane in other mechanism experiments. So we corrected to "more accumulated on the surface than in the cytosol of fungal cells".

I understand that the authors may have their theories but these need to be supported by data. Otherwise the authors can only speculate or guess.

A2) There was some confusion in our explanation of the results. So, we changed this description to “Both rhodamine-labeled peptides were more accumulated on the surface than in the cytosol of fungal cells (Figure 6). As shown in our report [17], pseudin-2 enters into the cytoplasm via itself-made pores and bound to nucleic acids. We suggested that P2-LZ1, -LZ2, -LZ3, and -LZ4 would have a similar mechanism of action to pseudin-2. On the other hand, P2-LZ5 was detected inside the cells, but not in the nucleus. This result suggests that the antifungal mechanisms of P2-LZ5 may be different to other peptides.”

Q3) In response to:

The 3D model was separated into figure 2.

This does not answer the request: Authors should display the CD spectra of peptides and liposomes alone.

This is important to understand how peptides change its structure upon membrane binding and to assess how reliable the models are.

A3) Following your comment, we have added the CD spectra of peptides in aqueous solution to the supporting information. We only tried to explain the difference between the facing parts of peptides in the 3D prediction, but we removed Fig. 3 in the manuscript because it could be misunderstood by readers.

Q4) In response to:

We understand the reviewer's opinion. But until now, we have never received a review requesting HPLC and MS data while submitting a paper. We confirmed the purity and molecular weight of the isolated peptide and performed an experiment.

I also understand the author's opinion but providing HPLC and MS data in manuscripts is a matter of good laboratory and publication practices. I cannot recommend publication unless the authors provide that information.

A4) We added the HPLC profile and Mass data of the peptides to supporting information. In fact, we started this study five years ago. So, some of the peptide data was only available as raw data. At the time of your request the last time, some of the peptides were all used and not in stock. Therefore, according to your request, some peptides were newly synthesized and data was added.

Q5) In response to: The number of fungal cells used in the antifungal activity assay and in the SEM experiment is different. So, we confirmed 4-times increment of the MIC values of each peptide in 1 × 106 cells/mL and we performed this experiment.

I can understand that, for some reason, authors decided to use a higher cell concentration and use a different MIC, but they need to explain why they did that. Why increase the cell concentration? If the MIC is increased, the authors should explain that they repeated the MIC assay and determined the MIC to be 4x the reported value.

A5) The fungal cells and peptide are incubated and then centrifuged several times during SEM preparation. At this time, the loss of the sample is severe. So, we increase the number of fungal cells and redefined the MIC values of peptides, resulted in 4 times MIC value of the peptide. It seems that the expression misunderstood you, and we changed it to "at the MIC value" in the method section.

Round 3

Reviewer 2 Report

The authors have addressed all my comments.